



# The Western United States MTBS-Interagency database of large wildfires, 1984–2024 (WUMI2024a)

A. Park Williams[1,2], Caroline S. Juang[3], Karen C. Short[4]

[1]Department of Geography; University of California, Los Angeles; Los Angeles, CA; USA
[2]Department of Atmospheric and Oceanic Sciences; University of California, Los Angeles; Los Angeles, CA; USA
[3]Department of Earth and Environmental Sciences; Columbia University; New York, NY; USA
[4]USDA Forest Service, Rocky Mountain Research Station, Missoula, MT, USA

*Correspondence to*: A. Park Williams (williams@geog.ucla.edu)

**Abstract.** Wildfire regimes of the western United States (US) have changed dramatically since the 1980s but our understanding
of the causes and effects of these changes is limited by a lack of a quality-controlled, publicly available database of wildfire
events that (1) spans from the 1980s to present, (2) represents wildfires across a wide range of sizes, and (3) includes mapped
fire perimeters and the area burned within. Here we present an updated and improved Western US MTBS-Interagency database
(WUMI2024a) of wildfire occurrences, perimeters, and burned-area maps, covering the period 1984–2024 and the geographic
domain of the 11 westernmost states in the contiguous US. The database represents 22,464 wildfires ≥1 km$^2$ in size, which we
compile by merging seven publicly available government databases. For over 46% of wildfires in our database (more than
10,300 wildfires), the maps of fire perimeters and area burned are based on 30-m satellite data provided by the US
government's Monitoring Trends in Burn Severity (MTBS) project, allowing our mapping and assessments of total area burned
to account for unburned areas within fire boundaries. For another 24% of fires, our database includes perimeter observations
provided by non-MTBS sources, meaning that only 30% of fire occurrences are without perimeter observations. For these fires
we tentatively assume perimeters are circular centred on the ignition location. Fires without perimeter observations are
relatively small and over 95% of area burned in the database is associated with fires with observed perimeters. The fire
perimeters and burned area maps will aid assessment of the landcover types that burn and can be used to improve simulations
of how historical fires have affected ecosystems and smoke emissions. The WUMI2024a can be quickly updated as new and
improved data become available. The WUMI2024a dataset and the code used to produce the dataset are available at
http://datadryad.org/share/Ox4oxdwdrhkmjUTpke7QgkfF--h-RLRbmMzGBhSmOr4 (Williams et al., 2025).

## 1 Introduction

The annual area burned by wildfires in the western United States (US) has increased dramatically over the past four decades,
largely due to rapid increases in the forested area burned by large wildfires (Juang et al., 2022). Fire activity is expected to
continue changing in the coming decades as complex and interactive responses to climate variability and change, changes in
vegetation, and human activities (Westerling et al., 2011; Bryant and Westerling, 2014; Keeley and Syphard, 2016; Parks et





al., 2016; McKenzie and Littell, 2017; Parks et al., 2018; Westerling, 2018; Hurteau et al., 2019; Abatzoglou et al., 2021). The effects of changing fire regimes on humans and ecosystems will also be complex, related not only to changes in the locations, sizes, and frequencies of fires, but also to their intensities, severities, and emissions of pollutants and greenhouse gases. Quantitative modeling is a critical tool for understanding the complex causes and effects of historical and future fire regimes

and assessing the effectiveness of strategies to avoid disastrous outcomes. However, the complexity of wildfires and their coupled interactions with ecosystems and human society prevent such model simulations from being performed across the large spatial scale of the western US without high degrees of parameterization. Instead, fire models that operate at regional to global scales are largely statistical, based on equations parameterized to optimally reproduce historical observations (Hantson et al., 2016; Williams and Abatzoglou, 2016).


Unfortunately, the usefulness of existing observational wildfire datasets for model parameterization and calibration are limited, even in the western US where government wildfire records are public, free, and extend back decades. Among the most valuable and heavily used observational datasets is a database of perimeters and high-resolution (30-m) satellite-based maps of burned area and fire severity developed by the interagency Monitoring Trends in Burn Severity (MTBS) program, covering 1984 to

near-present (Eidenshink et al., 2007). The main limitation of this dataset is that, in the western US, it only represents fires >4.04 km$^2$ in size, which account for most area burned in the region but only a small fraction of fire occurrences. Another commonly used dataset is the US Forest Service (USFS) Fire Program Analysis Fire Occurrence Dataset (FPA FOD), version 6, a database including ignition locations, dates, and sizes of over 2.3 million wildfires from 1992–2020, as reported by US federal, state, county, and local agencies (Short, 2022). This dataset includes fires of all sizes and was intensively quality

controlled to minimize duplicative fires, which are common in government records, but limitations are that it is point-based, doesn't span back into the 1980s or forward to near-present, and is subject to temporal changes and geographic inconsistencies in reporting practices by the fire service. Another interagency list of wildfire occurrences extends back through the 1980s but does not include fires reported by non-federal agencies and is not quality controlled to remove duplicates. In addition, the National Interagency Fire Center (NIFC) hosts a number of fire perimeter datasets on its Open Data site (https://data-

nific.opendata.arcgis.com/), but none is without a major caveat related to comprehensiveness, temporal coverage (the most comprehensive dataset covers <10 years), or lack of ignition dates or locations. A recent effort by the US Geological Survey (USGS) produced a quality-controlled compilation of US fire perimeters extending back to the 1800s (Welty and Jeffries, 2021), but limitations are that this dataset does not include fires that lack perimeter data and it currently ends in 2020.

Here we present the Western US MTBS-Interagency (WUMI) database of large (≥1 km$^2$) wildfires, version 2024a, during 1984–2024 (WUMI2024a). This version code reflects the final complete year represented in the dataset and accommodates the possibility of future releases before the dataset extends through 2025 or beyond. Previous versions of the WUMI, most recently Juang & Williams (2024), were not documented in a full-length paper and simply provided a list of wildfire events and monthly maps of western US area burned, and the MTBS dataset was the only source of actual fire perimeters and

footprints of area burned. All other fires were assumed circular. In this update, we merge and quality-control seven government databases of observed wildfire, five of which provide fire perimeters, to greatly increase the proportion of fires in the WUMI2024 with observed fire perimeters. In addition to the list of 22,464 wildfire start dates, start locations, and final sizes provided by the WUMI2024a, the database also now includes 1-km resolution maps of area burned as well as observed or estimated perimeters for each fire. The improved geographic representation of fire extent in this dataset allows for better

identification of the land-cover types burned in wildfire, which should improve the accuracy of statistical assessments of fire-landcover relationships across the large and ecologically diverse domain of the western US. The large and comprehensive archive of wildfire perimeters will also aid efforts to use high-resolution satellite imagery to map the severities of many more historical and recent fires than was previously possible, and improve accuracy of simulations of how historical wildfires have affected ecosystems, terrestrial carbon balance, and smoke emissions in the western US. Finally, the WUMI2024a can be

updated with relative ease as new and improved wildfire observations become available. This is critical, as continued changes in climate, ecosystems, and human activities are likely to promote continued occurrences of non-analog wildfire behavior that incentivize scientists to continue improving their quantitative models and understanding of how western US wildfire is likely to change in the coming decades.

## 2 Data sources and methods

The geographic domain of the WUMI2024a is the 11 westernmost states of the coterminous US: Arizona, California, Colorado, Idaho, Montana, New Mexico, Nevada, Oregon, Utah, Washington, and Wyoming. The time period covered is 1984–2024 and the WUMI2024a only represents wildfires ≥100 ha (1 km²) in size. The WUMI2024a consists of a list of all wildfire events in the database, monthly maps at 1-km resolution of area burned, and, for each event, a shapefile with the known or estimated fire perimeter and a map with 1-km resolution of gridded fraction of area burned. Here we describe each wildfire database we

used and then how we merged them to produce the WUMI2024a.

### 2.1 Data

Most datasets described below provide a fire start date but do not specify whether this is the date of ignition or discovery. We refer to the dates provided as "start dates." For fires with no reported ignition location (only perimeter), we provide the coordinates of the perimeter's centroid. When possible we replace these derived points with reported ignition locations from

agency reports (e.g., FPA FOD). Here we describe each data product and then we explain how they were integrated in section 2.2.

### 2.1.1 MTBS

The MTBS project (Eidenshink et al., 2007) maintains a database of Landsat-based maps of burned areas at 30-m resolution for large (>4.04 km²) fires spanning 1984–2024, though records for 2023 and 2024 were incomplete at the time of last access.

The MTBS dataset distinguishes four fire types: Wildfire, Prescribed fires, Wildland Fire Use, and Unknown. We exclude the Prescribed fire type. In addition to uses of the fire sizes provided as attributes in the MTBS records, which represent total area within the fire perimeter, we estimate the area burned by each fire as the summed area of all 30-m grid cells classified by MTBS as burned at low, moderate, or high severity. This provides a more accurate estimate of actual area burned by excluding areas classified as unburned. Data were accessed from https://www.mtbs.gov/ on April 26 2025.

**2.1.2 FPA FOD**

The USFS Fire Program Analysis Fire-Occurrence Database (FPA FOD), 6[th] edition (Short, 2022), is a compilation of final wildfire reports from the federal, state, and local fire services, currently spanning 1992–2020. The spatial component is limited to reported point locations of fire ignitions, although, when applicable, the database identifier for the corresponding MTBS perimeter is included for FPA FOD fires represented in the MTBS dataset. This dataset is described in detail by Short (2014).
Data were accessed from https://doi.org/10.2737/RDS-2013-0009.6 on November 4 2022.

**2.1.3 WFAIP**

The Wildland Fire Application Information Portal (WFAIP) includes an archive of point-based final fire reports from federal agencies, including the USFS and Fish and Wildlife Service from 1972–2020 and by the Bureau of Land Management, Bureau of Indian Affairs, Bureau of Reclamation, and National Park Service from 1972–2017. We limit incorporation of these records
to 1984–1991, because the FPA FOD dataset is sourced from the same federal reporting systems from 1992–2020, includes state and local data, and is scrubbed for redundancy as described in Short (2014). The WFAIP data were not used by the FPA FOD to extend that record prior to 1992 due to concerns about inconsistencies in reporting and the completeness of the records, but they are still a valuable resource for our purposes. Data were accessed from https://www.wildfire.gov/page/zip-files on December 15 2024.

**2.1.4 CalFire**

The California Department of Forestry and Fire Protection (CalFire) Resource Assessment Program (FRAP) maintains a database of perimeters for fire incidents in California. The first fire represented occurred in 1898 and the latest version includes records through 2024. The criteria that CalFire uses to determine which fires to include in the FRAP database have changed over time, but for the large $\geq 1$ km$^2$ fires and time period of focus in the WUMI2024a, the CalFire policy for inclusion has been
consistent. CalFire FRAP does not provide ignition locations, so we initially estimate each fire's ignition location as the central latitude and longitude within each fire's perimeter. Data were accessed from https://www.fire.ca.gov/what-we-do/fire-resource-assessment-program/fire-perimeters on April 26 2025.

**2.1.5 WFIGS**



The NIFC hosts a database of fire perimeters maintained by the Wildland Fire Interagency Geospatial Services (WFIGS)
Group. These perimeters are for wildland fire incidents that have been reported with an Integrated Reporting of Wildland Fire
Information identification code (IRWIN ID). The first fire in this database is from 2018 and the dataset is updated in real-time,
but federal records prior to 2021 are incomplete and perimeters may be missing for non-federal fires throughout the period of
record. This dataset only includes wildfires with perimeter data. Data were accessed from https://data-
nifc.opendata.arcgis.com/datasets/nifc::wfigs-interagency-fire-perimeters/about on April 26 2025.

**2.1.6 USGS**

The USGS compiled a dataset of US fire perimeters from 40 sources spanning from the 1800s through 2020 (Welty and
Jeffries, 2021). Their quality controlled "combined wildland fire polygon dataset" includes perimeters for >100,000 fires,
including >14,000 wildfires $\geq 1$ km$^2$ in the western US from 1984–2020. In the cases of multiple data products providing
overlapping fire boundaries, likely representing the same fire event, the authors dissolve the boundaries into a single best-
estimate boundary and they save the metadata reported by each product (e.g., fire name, ignition and/or discovery date, IRWIN
ID). In cases of multiple fire name listed for a given fire, we select the most common name, or the first name listed in the case
of a tie. For start date we use the ignition date when available and discovery date when not. In the case of multiple start dates
we use the one more commonly reported, and in the case of a tie we choose the earlier date. We discard the small proportion
(8%) of 1984–2020 western US fires $\geq 1$ km$^2$ with no start date information beyond the fire year. This dataset does not report
ignition locations and only includes fires with perimeter data. We therefore use this dataset to provide perimeter information
for fires listed by the more comprehensive fire-occurrence lists lacking perimeter data (FPA FOD and WFAIP), but not as a
primary          sourcde          of          fire-occurrence          records.          Data          were          accessed          from
https://www.sciencebase.gov/catalog/item/61aa537dd34eb622f699df81 on June 9 2025.

**2.1.7 IAFPH**

The NIFC hosts a database of US wildfire perimeters called the Interagency Fire Perimeter History (IAFPH), with some records
dating prior to the 1900s and spanning through 2023. There is a marked increase in the annual number of IAFPH fires
throughout the 20$^{th}$ century that reflects changes in availability of perimeter data rather than changes in actual fire frequency.
The IAFPH dataset only provides the year of fire occurrence, not the start date, and does not provide ignition location. For
these reasons the IAFPH is a useful resource for perimeters associated with known fires, but not as a primary source of fire-
occurrence          records.          Data          were          accessed          from          https://data-
nifc.opendata.arcgis.com/datasets/nifc::interagencyfireperimeterhistory-all-years-view/about on October 14 2024.

**2.2 Processing and merging of datasets**

**2.2.1 Within-dataset quality control**



From each dataset described above we remove fires <1 km² in area as well as duplicate entries in which multiple fires are
reported as occurring at the same location with the same size on the same date. We next remove likely duplicates, identified
as fires that occurred within 5 days, have matching names, and are within 200 km of each other (reported fire locations are
often imprecise or inaccurate). We have flexibility in searching for matching names, for example by allowing for variants such
as abbreviations (e.g., "MOUNTAIN" vs "MT" or "CANYON" vs "CYN") and uses of punctuation (e.g., "." or "#"). If a fire's
full name is within the name of another fire, we interpret them to be duplicates if they are within 5 days and 50 km. Among
pairs with an UNKNOWN name, we treat them as duplicates if they are within 25 km, 3 days, and 10% of the larger of the
fire sizes. If these criteria are met between a pair of fires but just one fire's name is UNKNOWN, they are considered duplicates
if they are within 1 day of each other. When fires determined to be duplicates have identical date and size, we keep the first
fire in the database. When date and size are not identical, we keep the fire with the earlier date and then the larger size. Finally,
for CalFire, WFIGS, IAFPH, and USGS, we identify pairs of fires with start dates within 14 days and overlapping perimeters.
If ≥50% of one fire's boundary is contained within the other's we only keep the larger fire. For WFAIP and FPA FOD, for
which boundaries are not available, we do the same but assume fires are circular centered on reported ignition location. After
automated quality control we visual inspect the fire lists and maps of fire perimeters to identify additional duplicates and errors
in fire dates or locations.

**2.2.2 Dataset merging**

To develop our final dataset of 1984–2024 large (≥1 km²) wildfires, we begin by making a master set of non-MTBS fires from
WFAIP, FPA FOD, WFIGS, and CalFire. We then merge this dataset with MTBS fires and then, when possible, replace
perimeters associated with WFAIP and FPA FOD fires, which we estimate to be circular, with fire perimeters from USGS and
IAFPH.

To produce the non-MTBS dataset, we merge CalFire with WFAIP for 1984–1991, FPA FOD for 1992–2020, and WFIGS for
2021–2024, prioritizing CalFire in cases of overlap. For WFIGS, most matches to CalFire records are easily made via common
IRWIN IDs. We additionally assume a non-CalFire fire to be a match to a given CalFire fire if it has the same name, has a
start-date within 5 days of the CalFire discovery date, and has an ignition location within 1.5° latitude and longitude of the
bounding box of the CalFire boundaries. For CalFire fires with UNKNOWN name and cases of non-matching names, we
consider a non-CalFire fire to be a match with, or part of, a CalFire fire if it is within 5 days of the CalFire fire and the ignition
location is within the CalFire bounding box. We additionally identify WFIGS fires as matches to CalFire fires if their listed
discovery dates are within 5 days and they have overlapping boundaries. When matches between non-CalFire and CalFire fires
are identified, the estimated CalFire ignition location (estimated as the centroid within the perimeter) is replaced with the
ignition location reported for the non-CalFire fire. Reported ignition locations outside the CalFire perimeter are adjusted to the
nearest point along the CalFire perimeter. Some CalFire fires represent fire complexes made of multiple fires that began



separately and merged. When multiple non-CalFire fires are linked to a given CalFire fire, we save this information and the CalFire ignition location is assigned the ignition location of the non-CalFire fire that is closest in size.

We next merge the non-MTBS dataset with MTBS by identifying matches between non-MTBS and MTBS fires as well as MTBS fires containing multiple smaller non-MTBS fires, prioritizing MTBS when possible due to the high-resolution (30 m) maps of area burned that MTBS provides. For WFIGS, we identify matches to MTBS by comparing IRWIN IDs. In addition, in producing the FPA FOD dataset, Short (2022) identified matches between FPA FOD and MTBS fires and provided the corresponding MTBS Fire IDs, which we use to link FPA FOD and MTBS fires. However, some matching fires between the FPA FOD and MTBS are not immediately evident based on the MTBS Fire IDs provided in the FPA FOD because the MTBS

undergoes regular revisions and some of its Fire IDs have changed. After using IRWIN IDs and MTBS Fire IDs to find matches between non-MTBS and MTBS fires, we identify additional matches using fire names, locations, and sizes, similar to our method described above to merge non-CalFire and CalFire fires. For the merge with MTBS our method is somewhat more liberal than for CalFire because, first, MTBS fires are exclusively large fires, and therefore more likely to represent fire complexes. Second, many MTBS burned areas are identified post-hoc with satellite imagery, but the pre- and post-fire images

used are generally not from satellite overpasses immediately before and after the fire so the burned areas attributed to some MTBS fires are actually from multiple fire events. To identify linkages between MTBS and non-MTBS fires, we first attempt to identify for each MTBS fire, any non-MTBS fires with the same name, dates within 14 days, and ignition locations within 1.5° longitude and latitude of the MTBS fire's bounding box. We next link non-MTBS and MTBS fires, regardless of name, if the start dates are within 5 days and the non-MTBS fire's ignition location is within the MTBS bounding box. Additionally,

we link WFIGS and CalFire fires to MTBS fires if the start dates are within 14 days and fire perimeters overlap. As with the CalFire merge, we allow for multiple non-MTBS fires to be linked to the same MTBS fire. Finally, we delete all MTBS fires that are (1) identified by MTBS as an Unknown fire type and (2) without any match to a non-MTBS fire, as these are likely to be prescribed fires that were never reported but produced burned areas detected post-hoc in the satellite imagery by the MTBS team. All remaining MTBS fires without linkages to non-MTBS fires are added to the list of non-MTBS fires to complete our

final list of WUMI2024a wildfire events.

At every step of the dataset merging process we conduct rigorous visual inspection of fire lists and perimeters to identify matching fires between datasets not identified by the automated process described above.

### 2.2.3 Supplementary datasets with fire perimeters

Although WFIGS is incomplete prior to 2021, USGS does not provide ignition locations or records of fires lacking perimeter data, and IAFPH does not provide ignition location or start date, these datasets can provide unique geographic information for many fires in the WUMI2024a. We therefore treat WFIGS during 2018–2020, USGS, and IAFPH as supplementary. During each supplementary dataset's period of overlap with the WUMI2024a (2018–2020 for WFIGS, 1984–2020 for USGS, 1984–



2023 for IAFPH), we identify matching fires in the WUMI2024a. Matches are identified when names match, fire starts are

within 14 days (or same year in the case of IAFPH), and the WUMI2024a ignition location is within 0.25° of the supplementary perimeter's bounding box. In cases of matching UNKNOWN fire names, the WUMI2024a ignition location must be within the supplementary fire's bounding box and its size must be within 10% of the supplementary fire's size. If a WUMI2024a fire is matched to more than one supplementary fire, WFIGS is prioritized because it has ignition location, then USGS because it has start date, and then IAFPH. In the case of a match to an FPA FOD or WFAIP fire, that fire is assigned the perimeter and

size of the supplementary fire and the ignition location is adjusted if outside the fire perimeter. In the case of a match between WFIGS and CalFire, the WFIGS ignition location is assigned to the CalFire fire and adjusted if outside the CalFire perimeter. Matches to other datasets (e.g., a supplementary fire to an MTBS fire) are recorded but no changes are made. Finally, for any FPA FOD or WFAIP fire not assigned a perimeter from an alternative dataset, we assume the fire perimeter to be circular centred on the ignition location.

## 2.3 Burned-area maps


For each fire in the WUMI2024a we produce a 1-km resolution map of fractional area burned. For MTBS fires, this is done by, for each 1-km grid cell in our study region, summing the area of all 30-m grid cells identified as burned in the original MTBS dataset. For all other fire perimeters, we assume the area within the fire perimeter is burned uniformly, accounting for sub-grid cell overlap with the fire perimeter near fire edges as well when fire perimeters indicate gaps within the area burned

or disconnected burned areas. Finally, as a quality control measure we make regional maps of monthly and annual area burned to identify cases of suspiciously high burned areas concentrated in time and space. This leads us to find duplicate fires not previously identified and, in some cases, use internet searches to identify incorrect dates or locations of fires. Over several iterations we correct found issues and remake the dataset following the methods described above.

## 2.4 Summary of the final dataset

The WUMI2024a database represents 22,228 wildfires ≥1 km$^2$ in size according to the area within the fire perimeter (some MTBS fires have less than 1 km$^2$ area burned according to the 30-m gridded classifications). However, 101 MTBS fires are composed of multiple non-MTBS fires. In these cases, we refer to the large MTBS fires as parent fires and the smaller fires within as sub-fires. Because these sub-fires generally have different ignition locations and dates from the parent fire, we produce an alternative list of WUMI2024a wildfire events that replaces each parent fire with its sub-fires. In this version, we

adjust the sizes of sub-fires so that they sum to equal the sizes of the parent fires. When the ignition location for a sub-fire is not within the satellite-derived area burned of the MTBS parent fire, we adjust the ignition location to the nearest burned location within the parent fire's footprint. In the alternative list of wildfire events that replaces each parent fire with its sub-fires, there are 22,464 wildfires ≥1 km$^2$ in size. Consideration of sub-fire events may be preferable over the parent events when the most accurate records of ignition dates, locations, and fire frequencies are needed. Consideration of the parent fires may

be preferable in applications requiring the most accurate maps area burned. In addition to maps of area burned by individual

fires, we provide monthly 1-km resolution maps of total area burned across the western US, prioritizing parent fires over sub-fires due to their superior spatial data.

Information on ignition source is available for many fires in the CalFire, FPA FOD, WFIGS, and USGS datasets. We preserve this information in the list of WUMI2024a wildfires. When one dataset (e.g., FPA FOD) provides ignition-source information for a given fire, but another dataset without ignition information is prioritized for that fire due to superior spatial information (e.g., MTBS), the ignition source information is carried over.

Figure 1 presents maps of the 22,464 ignition locations reported in the WUMI2024a as well as the areas burned. The map of
ignition locations (Fig. 1a) distinguishes fires ignited in forest versus non-forest areas and the grey contours in both maps bound forested areas. We assess forest using the 250-m resolution map of forest classifications from Ruefenacht et al. (2008). We consider an ignition to be in a forest area if its location lies in a 1-km grid cell for which ≥50% of the 250-m grid cells within are classified as forested. In the assessments below we divide the western US into four quadrant regions, mapped in Fig. 1: Pacific Northwest (PNW), Northern Rockies (N Rockies), California and Nevada (CA/NV), and Four Corners (4
Corners).

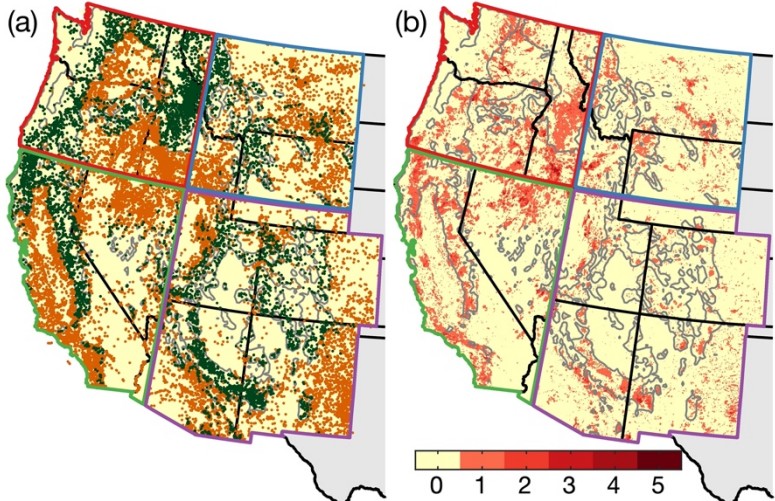

Figure 1. Maps of large (≥1 km$^2$) wildfires in the WUMI2024a from 1984–2024. (a) Ignition locations of 22,464 wildfires. Dot colors indicate locations outside (orange) versus inside (green) forested areas. (b) Number of times burned, approximated
as the sum of monthly 1-km resolution maps of fractional area burned. Grey contours bound forested areas where 12x12 km forest coverage is ≥50%. Western US domain is divided into four quadrant regions bounded by colors: (red) Pacific Northwest, (blue) Northern Rockies, (green) California/Nevada, and (purple) 4 Corners.

A unique feature of the WUMI2024a is its use of multiple data products to maximize the proportion of fire events for which
fire perimeter data are available. The top-left panel in Fig. 2 shows the annual number of fires for which perimeter data come



from each source (when multiple perimeters are available for the same fire, the prioritization order is MTBS, CalFire, WFIGS, USGS, IAFPH) as well as the number of fires assumed to be circular due to lack of perimeter data. We also provide annual maps of ignition locations that delineate the source of perimeter data in Supplemental Figure 1. Overall, 70% of fire events represented in the database include observed perimeters. The perimeters for two-thirds of these events (>46% of all events)

come from MTBS and thus are represented by especially accurate satellite-derived maps of area burned. The 30% of events without perimeter data are estimated to be circular. These are WFAIP or FPA FOD fires prior to 2021 that we could not link to fire perimeters from the other databases. Of the four quadrant regions shown in the Fig. 1 maps, the proportion of fires without perimeter data is the lowest (22%) in CA/NV due to the unique availability of fire perimeters from CalFire in California (Fig. 2, left side). In contrast, more than 41% of fires lack perimeter data in 4 Corners. Importantly, the fires lacking perimeter

observations are generally relatively small and account for less than 5% of total area burned in the WUMI2024a (Fig. 2, right side). In 4 Corners, the 41% of fires without perimeter observations account for less than 10% of area burned. In all regions, the proportion of fires lacking perimeter observations, as well as the burned areas represented by these fires, declined significantly from 1984 through 2020. In 2020, only 16% of all fires in the database lack perimeter observations and account for just 1.5% of area burned. The dramatic increase in the proportion of fires with perimeter data represents an encouraging

trend toward more rigorous collection and management of wildfire information since the 1980s, but is also illustrative of the unavoidable caveats related to dataset consistency, which all researchers should be aware of when using government records to analyze variability and trends in wildfire activity.

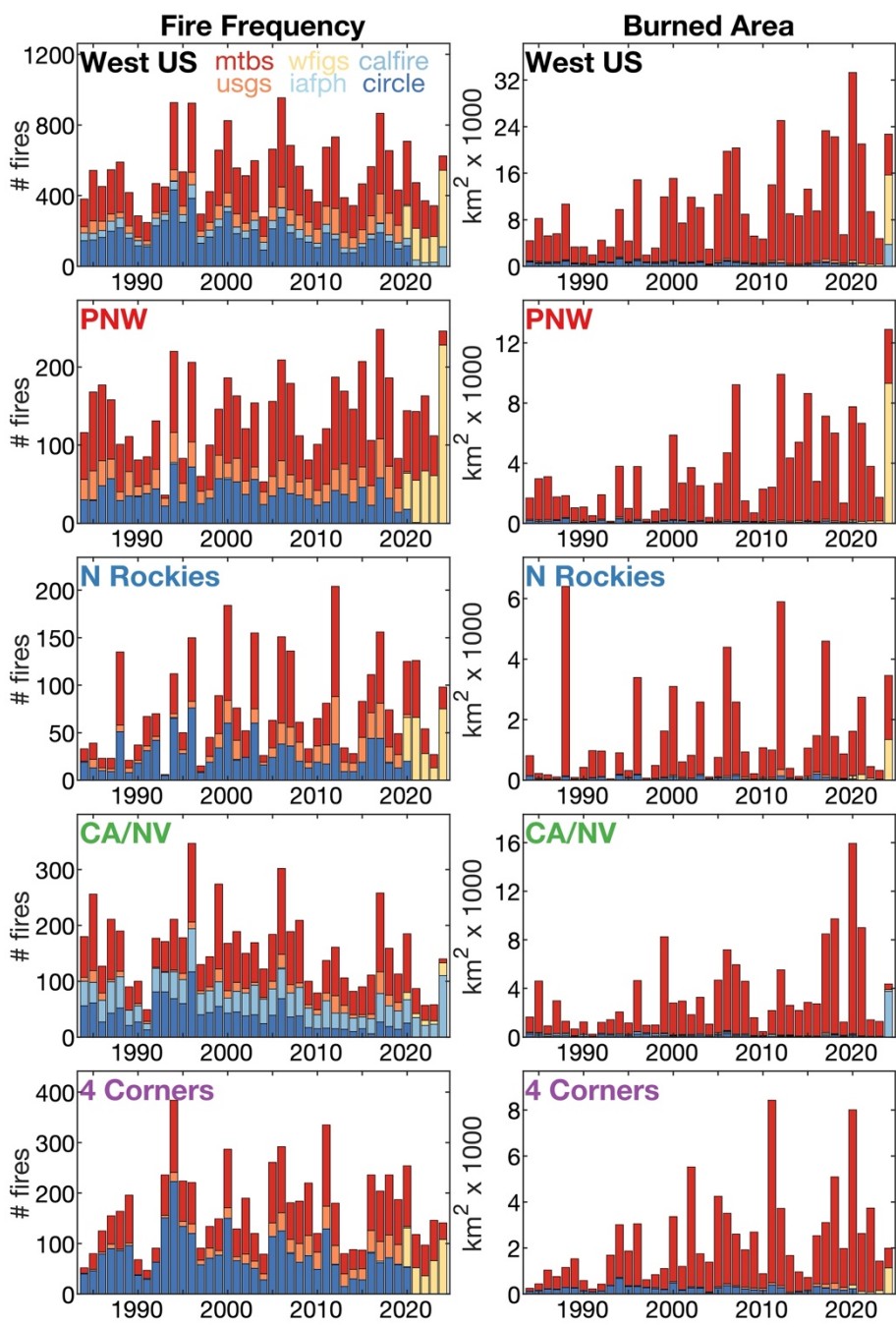

Figure 2. Dataset composition of WUMI2024a fire perimeters by year. (left panels) Annual number of wildfires with perimeter
data from MTBS, WFIGS, CalFire, USGS, or IAFPH, as well as the number of fires assumed circular due to lack of perimeter
data, for (top) the full western US and (lower rows) each of the four regions outlined in Fig. 1. (right panels) Same as left
panels except for area burned. Note that when perimeter data are available from more than one source, only one is used, first
prioritizing MTBS, then CalFire, then WFIGS, then USGS, then IAFPH.

## 3 Data quality and dataset intercomparison

Efforts to assess spatiotemporal variations and changes in historical fire activity are challenged by inconsistencies in the

methods to collect, report, and store wildfire information across time and space. Among government records, the types of

information recorded, or whether a record for a given fire is made at all, have been inconsistent over time as well as across

agencies, and the same fire event is often recorded by multiple government agencies, but often with differing names, dates, or

sizes. Many consistency issues have been alleviated through the Integrated Reporting of Wildland Fire Information (IRWIN)

initiative, but this is a relatively new initiative that has been increasingly implemented since 2014

(https://www.wildfire.gov/application/irwin-integrated-reporting-wildfire-information). It is also important to acknowledge

that the government wildfire records used for the WUMI2024a are more appropriately viewed as records of fire suppression

efforts than as records of all fire occurrences. Thus, the WUMI2024a probably underrepresents the occurrences of very small

or remote wildfires, as these fires may be more likely to never command attention from an official fire-suppression agency, or

to require such a minor suppression effort as to have not been recorded. Further, the data associated with relatively small fires

that require fewer suppression resources and are of relatively little consequence may be more likely to receive less scrutiny for

accuracy than lager fires, and policies for whether or how data are recorded for these fires are more likely to change over time

than for larger fires. For example, the FPA FOD dataset exhibits irregularities in the frequencies of very small (e.g., <4 ha)

fires reported by non-federal entities in developed areas that are likely artifacts of changes in reporting levels from local fire

departments (Jorge et al., 2025). Given that fire occurrences are dominated by very small events, assessments of fire frequency

are especially vulnerable to errors or inconsistencies in the reporting of small fires, thus motivating our choice to limit the

WUMI2024a to larger fires ≥100 ha in size. While satellite imagery can be used to supplement and improve the accuracy of

government records of fire incidents, which we do through use of the Landsat-derived maps of area burned by the very large

fires represented by MTBS, no publicly available database of satellite imagery exists that can be used to systematically identify

the dates, start locations, and spatial extents of western US wildfires from the mid-1980s to present.

The most substantial temporal inconsistencies in the accuracy of the WUMI2024a are likely caused by the shift from reliance

on the WFAIP as the primary source of fire-occurrence information prior to 1992 to the FPA FOD from 1992–2020. Indeed,

by comparing time series of the annual fire frequencies derived from the WFAIP and FPA FOD datasets during their period

of overlap, we see that the WFAIP list of federal incidents consistently represents fewer wildfires than the more comprehensive

FPA FOD list of incidents reported by federal, state, and local agencies (Fig. 3). Comparing these datasets during their first 10

years of overlap (1992–2001), and focusing on the large ≥100 ha fires of focus in the WUMI2024a, we find that WFAIP fires

are 21% less numerous than FPA FOD fires. Essentially all of this offset is explained by the inclusion of data from state,

county, and local (ST/C&L) agencies in the FPA FOD dataset. The orange dashed time series in Fig. 3 represent an alternative

version of the FPA FOD dataset that excludes ST/C&L fires, and shows that excluding ST/C&L fires from FPA FOD causes

much better alignment with WFAIP during 1992–2001. Assessing fires ignited in forest versus non-forest separately, and

within each of the four regions visualized in Fig. 1, it is clarified in Fig. 3 the under-representation of WFAIP fires is mostly

confined to the non-forest areas of the southwest US, particularly in 4 Corners but also in CA/NV. In non-forested areas of 4

Corners, approximately 45% of all FPA FOD fires ≥100 ha in size are reported by ST/C&L agencies, and thus the WUMI2024a

probably underrepresents the actual frequency of non-forest fires in this region by approximately half prior to 1992, relative

to what may be expected if the FPA FOD data were available during this time. In the California portion of CA/NV, where

ST/C&L reports are also relatively common in the FPA FOD, the negative bias in pre-1992 fire frequencies is largely addressed

in the WUMI2024a through the use of CalFire data, which do include fires addressed by non-federal agencies (as indicated by

pre-1992 non-forest fires in the California/Nevada region being considerably more frequent in WUMI2024a than in WFAIP).

Under-representation of pre-1992 fires may be most extreme in the easternmost states of the western US (Montana, Wyoming,

Colorado, and New Mexico). In particular, the maps of annual fire occurrences provided in Supplementary Figure 1 indicate

a complete lack of fires in 1984 in the states of Wyoming and New Mexico, which is almost certainly an artifact of inconsistent

reporting practices.

Importantly, Fig. 4 shows that the bias toward too few pre-1992 fires is less consequential to assessments of annual area burned

than to fire frequency, as indicated by a much better match between the FPA FOD and WFAIP time series of area burned

during their period of overlap. This is because the ST/C&L fires missing from WFAIP tend to be relatively small, whereas

regional time series of area burned are dominated by a smaller proportion of very large fires that command federal responses.

Users who prefer a dataset built from a more temporally consistent set of data sources may consider excluding fires reported

by ST/C&L agencies. Overall, the WUMI2024a underrepresents fire frequencies in 1984–1991 relative to 1992–2020, but this effect appears isolated mostly to relatively small fires in non-forest areas, particularly in the easternmost states of the western US. We therefore caution against over-interpretation of the relatively low frequency of pre-1992 fires, particularly small non-forest fires, as represented by WUMI2024a or other databases that extend back to the 1980s or prior.

There are probably additional artifacts associated with the 2021 transition from reliance on FPA FOD to WFIGS as the primary source of wildfire occurrence data. While WFIGS does include fires reported by non-federal agencies, fires are not included in WFIGS if strict reporting standards are not met. The effects of this shift are difficult to readily assess because WFIGS is not complete during its 2018–2020 period of overlap with FPA FOD. Some insights may be gleaned, however, through comparison to the independent satellite-based FIRED dataset (Balch et al., 2020), a delineation of the gridded MODIS v6.1 burned area

product (Giglio et al., 2016) into individual fire events. The blue dashed time series in Fig. 3 show MODIS-derived annual fire frequencies, excluding fires <1 km$^2$ in size (assuming burned 500-m grid cells burn uniformly) as well as fires in heavily urbanized or agricultural areas where satellite fire detections are less likely represent wildfires. Comparing MODIS to WUMI2024a in non-forest areas, we see WUMI2024a fire frequencies tend to be somewhat higher than those derived from MODIS in 2001–2020, but not during 2021–2023 (Fig. 3, right side). This suggests that WFGIS may systematically

underrepresent fire frequencies relative to what would be expected if FPA FOD was available following 2020. In forest areas, on the other hand, MODIS fires are consistently more frequent than indicated by WUMI2024a (Fig. 3, left side). The higher forest-fire frequencies in MODIS are most prominent in PNW and N Rockies, but also emerge in CA/NV and 4 Corners after 2017. However, there is not a clear post-2020 shift in WUMI2024a fire frequencies relative to those calculated from MODIS as there is for non-forest fire, and it is therefore unclear how the shift from FPA FOD to WFIGS may affect accuracy of post-

2020 forest-fire frequencies.

The higher frequencies of forest fires in the MODIS FIRED dataset relative to WUMI2024a are interesting. One likely cause is that government records often attribute geographically separated burned areas to the same event, but the FIRED dataset may more often delineate these as distinct events. In addition, the MODIS burned area dataset does not distinguish prescribed fires



from wildfires and may also capture wildfires in remote areas that are small and short-lived enough to not be managed by a

federal or local agency. Importantly, MODIS records of annual burned area are in tight agreement with the WUMI2024a (Fig.

4) despite the notable disagreements in fire frequency discussed above, reinforcing our interpretation that assessments of

spatiotemporal variations in areas burned are far less sensitive than fire frequencies to historical inconsistencies in wildfire

reporting.




Figure 3. Annual frequency of large wildfires, by dataset. Panels on left and right represent fires ignited in forest and non-forest areas, respectively (see Fig. 1a). Top row represents the full western US study region and rows below represent the four quadrant regions mapped in Fig. 1. Colors distinguish the databases used: WUMI2024a (bold black), WFAIP (light green), FPA FOD (orange), and WFIGS (light purple). The dashed orange line is an alternative version of FPA FOD that excludes fire reports from non-federal agencies. The dashed blue time series is calculated from the independent MODIS FIRED dataset of satellite-derived fire events for 2001–2023 (excluding fires centered on 1-km grid cells where agricultural and/or urban coverage exceeds 75% according to the USGS National Land Cover Dataset). The WFAIP time series ends in 2017 because


this is when its records from most federal agencies ends. The WFIGS time series begins in 2021 because it is not complete
prior to that year.



Figure 4. As in Figure 3 but for annual area burned. Forest area burned by each fire is estimated by multiplying the 1-km
resolution maps of area burned and fractional forest coverage (Ruefenacht et al., 2008) and summing.


## 4 Data availability

The WUMI2024a dataset and the code used to produce the dataset are available for peer review at http://datadryad.org/share/Ox4oxdwdrhkmjUTpke7QgkfF--h-RLRbmMzGBhSmOr4 (Williams et al., 2025).

## 5 Conclusions

The WUMI2024a is a comprehensive and quality controlled database of more than 22,000 large ($\geq$1 km$^2$) wildfires in the western US, currently covering 1984–2024. The database includes start dates, final sizes, and digital maps of perimeters and area burned for each fire. It can be updated with relative ease as new and improved data become available. We compiled this database from seven government datasets and this is the only such compilation to cover the full period of 1984 to 2024 and include fires as small as 1 km$^2$. For more than 10,300 (~46%) of fires represented, the maps of area burned are calculated from

high-resolution satellite imagery and thus account for islands of unburned areas within fire perimeters, which are important to ecosystem recovery and resilience (Coop et al., 2019; Krawchuk et al., 2020). For another 24% of fires we provide fire perimeters accessed from other government datasets, leaving us to resort to an assumption of circular perimeters for just 30% of fires, which account for only 5% of the area burned. Our primary motives in producing this database include (1) updated assessments of trends and extremes in the wildfire frequency and area burned; (2) improved potential for statistical analyses

and modelling of wildfire activity; (3) linkage of our fire perimeters to independent observations of fire progression (e.g, Balch et al., 2020; Lizundia-Loiola et al., 2021), area burned (e.g., Hawbaker et al., 2017), fire severity (e.g., He et al., 2023), and hydrology (e.g., Williams et al., 2022); and, ultimately, (4) improved ability to simulate historical and ongoing effects of wildfire on ecosystems, terrestrial carbon, smoke, hydrology, and society.

Importantly, no historical wildfire database is without major caveats associated with temporal and geographic inconsistencies in the practices used to collect, archive, or disseminate data over time. Satellite-derived observations of fire offer promising opportunities to sidestep the biases and methodological uncertainties inherent to the government records we used to develop the WUMI2024a, but the temporal frequencies of publicly available satellite products do not allow for confident detection of fire-start timing or location across the western US prior to the 2000s, and moderate-to-low spatial resolution inhibits accurate

detection of small fires and fine-scale features of burned-area boundaries. While Landsat imagery is invaluable for production

of accurate maps of area burned, which we capitalize on through the MTBS dataset, Landsat overpass times are too infrequent (16 days) to detect many important wildfire features such as the start date and location. Thus, while Landsat imagery can be used to produce wall-to-wall maps of area burned with relatively low (e.g., annual) temporal resolution (Hawbaker et al., 2017), additional observational datasets are necessary to confidently associate Landsat images of burned area with a start date

or ignition location. Further, the pre- and post-fire imagery used to identify burned areas may be obscured by clouds, gaps in the imagery data, or changes in vegetation characteristics. Nonetheless, the value of the WUMI2024a can be improved by building beyond the large-fire focus of the MTBS to develop 30-m resolution maps of area burned for fires with known perimeters and map the actual perimeters and areas burned for fires with unknown perimeters. Landsat imagery should also be used to identify the footprints and general timing (e.g., year, season, or month) of fire events not yet represented in our database,

or identify errors in the locations or dates of fires that are represented, which can guide efforts to expand and improve future versions of the WUMI database.

## 6 Author contributions

APW, CSJ, and KCS developed the methodology. APW carried them out, performed the analyses, and prepared the manuscript. CSJ and KCS provided edits to the manuscript.

## 7 Competing interests

The authors declare that they have no conflict of interest.

## 8 Acknowledgements

This research was supported by the Zegar Family Foundation, the John D. and Catherine T. MacArthur Foundation, the Gordon

and Betty Moore Foundation (11974 and 13283), the UCLA Sustainable LA Grand Challenge, and the USGS Southwestern Climate Adaptation Science Center (G24AC00611 and G24AC00080).



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
