# Peer review of "The Western United States MTBS-Interagency database of large wildfires, 1984–2024 (WUMI2024a)"

_Earth System Science Data, 2025_

## Author Comment (AC1)

**Reviewer 1 general comments**

The objective of the paper is very interesting and extremely useful. Although remote sensing increasingly plays and important role in monitoring the earth surface, and in particular wildfire activity, it lacks the temporal extent to bring the needed confidence of results that assess the role and interaction of fire in the earth systems. This paper addresses this gap by merging different datasets to build a comprehensive and consistent geo-database of wildfire activity over western US. The paper reads well and is clear. In terms of presentation, the paper could benefit from a visual diagram, in the methods section, on how the datasets interact and the rules (temporal and spatial) applied to remove duplicate fire records. In addition, for a paper on a new compiled dataset, it would benefit presenting all the attributes in a table, including also the definition/characteristics that are contained in dataset attributes. This not only makes it clear what the measurements and classification are, but also quickly informs potential users on the fitfor-purpose of the dataset is for their application.

We thank the reviewer for their thorough and constructive review of our paper. We address each comment below (in blue font). In addressing the reviews we discovered some minor errors in our code and also performed additional manual inspections of the fire lists and burned area maps that led to a revised dataset with approximately 1% fewer wildfires. This revision did not cause notable changes to the regional patterns of fire frequency or burned area. The paper and the WUMI2024a database archived online have been revised to reflect these changes.

In response to the suggestions made in the general comments above:

Diagram: In response to this recommendation as well as a similar recommendation from Reviewer #2 we have now added a flow chart (new Fig. 1, and pasted below) that visualizes the process of data preparation, quality control, and inter-dataset merging. We feel that the flow chart is most effective if the details it provides remain at a relatively high level and we continue to use the main text to describe the specific rules used to identify duplicate fires within and between datasets are best left in the methods text.

Figure 1. Flowchart outlining the approach to develop the WUMI2024a.

Attribute table: We agree that a table listing and describing the attributes of the dataset would be helpful. We have added a table to the Supplement (also provided below) and reference the table in section 2.4 (Summary of the final dataset).

Table S1. Attribute names (left) and descriptions (right) provided in the lists of WUMI2024a wildfire events.

| fireid                 | Unique fire identifier indicating date (YYYYMMDD) followed by latitude in degrees north times 10000 followed by longitude in degrees west times 10000                                                                                                               |
|------------------------|---------------------------------------------------------------------------------------------------------------------------------------------------------------------------------------------------------------------------------------------------------------------|
| dataset                | Name of the dataset used to retrieve the fire name, start date, and ignition location (mtbs, calfire, wfigs, fpa fod, or wfaip)                                                                                                                                     |
| agency                 | Agency reporting the fire (not available for mtbs fires)                                                                                                                                                                                                            |
| name                   | Fire name                                                                                                                                                                                                                                                           |
| year                   | Year of fire start                                                                                                                                                                                                                                                  |
| month                  | Month of fire start                                                                                                                                                                                                                                                 |
| day                    | Day of month of fire start                                                                                                                                                                                                                                          |
| lat                    | Latitude of ignition                                                                                                                                                                                                                                                |
| lon                    | Longitude of ignition                                                                                                                                                                                                                                               |
| lat_ll                 | Minimum latitude of fire perimeter (if fire perimeter is available)                                                                                                                                                                                                 |
| lon_ll                 | Minimum longitude of fire perimeter (if fire perimeter is available)                                                                                                                                                                                                |
| lat_ur                 | Maximum latitude of fire perimeter (if fire perimeter is available)                                                                                                                                                                                                 |
| lon_ur                 | Maximum longitude of fire perimeter (if fire perimeter is available)                                                                                                                                                                                                |
| poly_area_ha           | Reported fire size in hectares (within perimeter area if fire perimeter is available)                                                                                                                                                                               |
| burn_area_ha           | Area burned (actual area burned within the perimeter if an mtbs fire or a subfire of an MTBS fire with an adjusted area burned)                                                                                                                                     |
| mtbs_name              | Name of the mtbs fire (for fires either in the mtbs datast or non-mtbs subfires that are part of an mtbs parent fire)                                                                                                                                               |
| mtbs_ID                | Identification code the mtbs fire (for fires either in the mtbs datast or non-mtbs subfires that are part of an mtbs parent fire)                                                                                                                                   |
| irwinid                | Integrated Reporting of Wildfire Inormation (IRWIN) Identification code                                                                                                                                                                                             |
| FOD_ID                 | Fire Occurrence Dataset identification codes for fpa_fod fires                                                                                                                                                                                                      |
| FPA_ID                 | Fire Program Analysis identification codes for fpa_fod fires                                                                                                                                                                                                        |
| object_ID_wfaip        | If a wfaip fire, the identification number that can be cross-referenced to the object_id attribute in the original list of wfaip fires provided in fire lists/wfaip/wfaip fires qc.txt                                                                              |
| object_ID_fpafod       | If an fpa fod fire, the identification number that can be cross-referenced to the object_id_attribute in the original list of fpa_fod fires provided in fire lists/fpa_fod/fpa_fod fires_qc.txt                                                                     |
| object_ID_wfigs        | If a wfigs fire, or an fpa_fod fire that was matched to a wfigs fire, the identification number that can be cross-referenced to the object_id attribute in the original list of wfigs fires provided in fire lists/wfigs/wfigs fires qc.txt                         |
| object_ID_calfire      | If a calfire fire, or a non-calfire fire that was matched to or part of a calfire fire, the identification number that can be cross-referenced to the object id attribute in the original list of calfire fires provided in fire lists/calfire/calfire fires qc.txt |
| object_ID_usgs         | If matched to a usgs fire, the identification number that can be cross-referenced to the object_id attribute in the original list of calfire fires provided in fire_lists/usgs/usgs_fires_qc.txt                                                                    |
| object_ID_iafph        | If matched to an iafph fire, the identification number that can be cross-referenced to the object_id attribute in the original list of calfire fires provided in fire lists/usgs/usgs fires qc.txt                                                                  |
| cause_human_or_natural | General ignition cause if available, HUMAN or NATURAL                                                                                                                                                                                                               |
| cause_specific         | Specific ignition cause if available                                                                                                                                                                                                                                |

**specific comments**

The purpose of the paper is achieved but the work is not of exceptional quality and compleness. I was expecting that for such an important topic - where one needs to merge old and new and different information types – that the new dataset and its merging methodology would set a standard on how it could be done with the perspective that it would be regularly updated with attributes that are of important to fire managers, ecologist, fire ecologist and climate researchers, to name a few. Attributes like end-date, fire spread rate, intensity, power, landcover, and fragmentation, can be retrieved and added, especially to recent records. The dataset, although very useful, is limited to what is commonly recorded by the different datasets and struggles to show added value in terms of new information useful for different applications

We agree with the reviewer that there are many opportunities to deepen the dataset by providing additional attributes for the fires represented in our database and we have heavily revised the final paragraph of the Conclusions section to better describe how the WUMI2024a can and should be deepened.

As the reviewer implies, there are many variables that one could consider, and each would add value to the database, but also each would come with its own caveats and would warrant unique attention from peer reviewers. The current version of the WUMI2024a was not developed to serve as an end-all data clearinghouse for all-things fire, but instead as the most comprehensive database available for the start dates, locations, and, when possible, perimeters for wildfires ≥1 km² in the western US from the mid-1980s to near-present. We are confident that this database, in its current form, will provide unique value beyond any database of fire occurrences and perimeters developed for the region to date, including by easing the work required on future efforts to align western US wildfire events with information from other databases related to fire attributes and effects.

**The new Conclusions paragraph reads (L511-526):**

"Finally, the value of the WUMI2024a can be improved upon by expanding it to include more than wildfire start dates, locations, causes, perimeters, and 1-km resolution maps of area burned. For example, many fires in the WUMI2024a can be linked to those represented in the ICS-209-PLUS database (St. Denis et al., 2023) to provide information about fire management costs, personnel, and impacts on people and property. Additionally, Landsat imagery can be used to improve our maps of area burned for fires in the WUMI2024a that are not represented among the very large >4.04 km2 fires mapped by the MTBS. Landsat imagery can also be used to detect the perimeters and areas burned for fires with unknown perimeters that we currently assume to be circular, identify additional fire events not yet represented in our database, and expand beyond our maps of burned area to also develop high-resolution maps of fire severity (e.g., Parks et al., 2019) for all fires in the database. Likewise, the MODIS-based FIRED dataset (Balch et al., 2020) can be used to decompose the fire maps in our database into maps of daily fire progression, and capabilities for mapping of fire spread are deepening further through use of the Visible Infrared Imaging Radiometer Suite (VIIRS) instrument (Schroeder et al., 2014; Chen et al., 2022). Imagery from MODIS and VIIRS can also be used to map fire intensity via retrievals of fire radiative power (Schroeder et al., 2014; Giglio et al., 2016). While the WUMI2024a database of wildfire events has been designed to accommodate future updates and expansions, including additional metrics related to fire processes (e.g., spread) and impacts (e.g., severity), its current form advances beyond currently available databases of western US wildfire events and will be a robust and valuable resource for researchers and practitioners in the field of wildfire science."

The authors, although focusing on achieving the higher quality records for the dataset, do not supply a quality/confidence indicator associated with each record. Meaning that users that may want to screen the dataset to remove data that could be uncertain. The compiled dataset should offer a confidence layer. It that regard, the paper fails to set a standard for dataset compilation, and it comes across that quality is reduced to removing duplicates. The authors should consider developing a confidence indicator.

We thank the reviewer for this comment, while we have not added fire-specific confidence indicators, we have revised the paper to clarify our advice as to how the fire data from various datasets should be prioritized. We do not develop and implement an official confidence scoring system because all of the fire datasets we use are observational, and thus theoretically true. Of course no dataset is perfect, thus the reason why a confidence indicator would be nice, but this cuts both ways. Because all of the datasets we use are not only imperfect, but also incomplete in terms of fire size, temporal coverage, and/or availability of perimeter information. In addition, all datasets, possibly with the exception of CalFire, are subject to temporal instabilities due to changes in the type of information that they incorporate during their periods of coverage. This all contributes to our determination that there is no ideal 'ground truth' dataset against which to benchmark our dataset in terms of occurrence probability, size, start-date, or perimeter, and if such a benchmark dataset did exist, then we'd just use that dataset for our research rather than put forth such an effort to develop our own dataset.

Although we chose to not add confidence indicators to the dataset, this comment as well as a suggestion from Reviewer #2 did encourage us to add a new paragraph to the Conclusions section about limitations of our dataset and associated recommendations for use. That new text is (L489-509):

"The WUMI2024a also has caveats. First, it relies on publicly available government records of wildfire incidents, which can be more accurately characterized as records of fire response and management rather than as purely records of wildfire occurrence. This means the WUMI2024a is missing any wildfires not recorded by a fire management agency and incorporated into a publicly available database. For example, some wildfires may have never been detected and reported, and others may have been extinguished without receiving official attention from a fire management agency. Thus, the database's completeness and accuracy are subject to temporal and geographic inconsistencies related to changes in practices related to fire detection, management, and data archival. These limitations are well exemplified by the fact that the WUMI2024a indicates zero wildfires in Wyoming and New Mexico in 1984 (Fig. S2). The lack of large 1984 wildfires in two western US states is likely an artifact and we suggest excluding 1984 in assessments that include the interior Western US. We further presented evidence that fire frequencies in the WUMI2024a are highly likely to be artificially low prior to 1992 (the first year of the FPA FOD) in non-forested areas of the interior West because many wildfires in these areas are addressed by non-federal agencies (ST/C&L), which are not represented in the WFAIP database (Figs. 4, S3a). The negative biases in pre-1992 frequencies are likely to be largest in the non-forest areas of New Mexico, Arizona, Colorado, and Wyoming, as these are the areas where ST/C&L fires are most common in the FPA FOD, so we caution against analyses that rely on consistent fire reporting from the 1980s through 1990s in these areas. Our database does comprehensively represent non-federal fires in California, on the other hand, via data from CalFire. Finally, we were unable to find perimeter data for 29% of the wildfire events in the WUMI2024a. While these fires are relatively small, only accounting for less than 5% of area burned in the dataset, and we do adjust the circular perimeters and maps of burned area to exclude open water, ice, and barren ground, we warn against use of the circular fire features for applications that require accurate maps of area burned. These derived features are provided as

resources for applications where an approximated map of area burned is better than none at all (e.g., as inputs for some hydrological or smoke-emission modelling exercises), but should be used with caution."

The authors used several rules to remove duplicate records when merging the different datasets. These rules are mostly based on chosen thresholds in space and time. The authors should state what is the rationale behind these, and why the chosen values.

In response to this comment as well as a comment from Reviewer #2 we have added additional text to sections 2.2.1 (Within-dataset quality control) and 2.2.2 (Dataset merging).

The most substantive addition is to the end of section 2.2.2, where we now better explain that no one-size-fits-all set of rules can feasibly be used to identify and fix all of the issues related to duplicate fires that appear within and across datasets, and regardless of the rules we put in place, it was unavoidable that intensive manual work was necessary to supplement the automated process. We have therefore expanded on our previous statement at the end of 2.2.2 about the role of our visual inspections of the data (L224-232):

"Importantly, we found it infeasible to automate a one-size-fits-all set of rules that effectively detects duplicate fires within or between datasets without need for intensive additional scrutiny. For example, the same fire is often assigned different names by different government agencies; the same name may be spelled or misspelled in a range of ways; and dates, locations, and sizes are not always reported consistently across agencies, sometimes with large differences. Therefore, at every step of the quality-control and dataset-merging process we conduct rigorous visual inspections of fire lists and perimeter maps to identify additional duplicate fires between or within datasets as well as fires that were misidentified as redundant by our automated routine. Due to the intensive nature of the visual inspection and manual correction process we are confident that the quality of the final WUMI2024a dataset, while certainly not without remaining errors, is not highly sensitive to the specific rules implemented in the automated portion of the process."

Our addition to 2.2.1 is specific to the 200-km and 5-day thresholds that Reviewer #1 asks about later in this review. We describe that revision later in this document in response to relevant reviewer question.

Circular areas are never representative of fire scars. Over flat grasslands under constant conditions burn scars can appear ellipsoidal, but over other landscapes they are rarely circular. By representing them as such, the authors risk overlaying a fire scar over bare soil or water, requiring users to treat the data prior to using it when landcover is important. This kind of inconsistencies should be avoided.

We thank the reviewer for this comment, which pertains to our use of circular fire perimeters symmetrical about the ignition point in cases of fires only represented in the FPA FOD or WFAIP databases, as these databases only include ignition location and fire size, but no information on shape. In response to this comment we have revised our approach by adjusting the shapes and sizes of circular perimeters to no longer allow for burned area in areas defined by the National Land Cover Database as open water or bare soil/rock/ice. In doing so, we have improved the realism of our estimated maps of perimeters and area burned for these fires by no longer assuming that fire burned in areas that are unable to burn in reality, but we retain the general approach of assuming a generally circular fire shape when no observations of fire shape are available from MTBS, CalFire, USGS, or IAFPH.

For context, we remind the reviewer that the circular-fire assumption is only made for the minority of fires, most relatively small, for which we do not have fire-perimeter data, which account for 29% of fire events and 4.5% of burned area in our database. Without using remotely sensed imagery to detect the true perimeters of these events ourselves, which we think would be a worthy but time- and resource-intensive effort that deserves a data release and method-intensive paper of its own, we must make an approximation if we wish to represent these fires in our maps of area burned.

We prefer to include these perimeter-less fires in our maps of area burned rather than leave them unmapped because, while surely inaccurate at fine spatial scales, including these fires improves the accuracy of our maps of area burned at broader, regional scales. This aids the accuracy of our dataset for assessments of temporal variations and trends in area burned and will improve the usefulness of our dataset for a range of applications, such as forcing models that simulate vegetation ecosystems, smoke/carbon emissions, or hydrology. Users of our perimeter and areaburned maps are of course free to disregard fires that we represent as circular by, if using our fire-specific maps, simply not using the data for these fires. If using our gridded maps of monthly area burned for the western US, one can easily subtract away the values from the gridded maps of area burned associated with circular fires.

To help users avoid placing undue trust in the circular fire perimeters we included the following disclaimer within the new Conclusions paragraph that we quoted in response to another reviewer comment above. The disclaimer reads (L504-509):

"Finally, we were unable to find perimeter data for 29% of the wildfire events in the WUMI2024a. While these fires are relatively small, only accounting for less than 5% of area burned in the dataset, and we do adjust the circular perimeters and maps of burned area to exclude open water, ice, and barren ground, we warn against use of the circular fire features for applications that require accurate maps of area burned. These derived features are provided as resources for applications where an approximated map of area burned is better than none at all (e.g., as inputs for some hydrological or smoke-emission modelling exercises), but should be used with caution."

**technical corrections**

Line 37: sentence could finish with a reference to a study supporting the statement.

This sentence is the first of a two-sentence sequence that, together, make the point that fire and it's coupled interactions with vegetation and humans are too complex to be modeled in a purely dynamic, process-based fashion when it comes to simulations across large spatial scales like the western US, and thus the models used for simulations across regional to global scales are largely statistical. The relevant references are provided at the end of the second sentence (L37-41):

"However, the complexity of wildfires and their coupled interactions with ecosystems and human society prevent such model simulations from being performed across the large spatial scale of the western US without high degrees of parameterization. Instead, fire models that operate at regional to global scales are largely statistical, based on equations parameterized to optimally reproduce historical observations in wildfire activity (Hantson et al., 2016; Williams and Abatzoglou, 2016)."

Line 83-84: this sentence can be confusing, I recommend reminding that the datasets includes fires that are lower than 1km and that proportion of BA is what is recorded in at every 1km2 gridcell.

We have revised this sentence to more clearly specify that when a fire only burns a portion of a 1-km grid cell, then our 1-km maps of burned area indicate the fraction of the grid cell that burned. The revised sentence reads (L84-88):

"The WUMI2024a consists of a list of all wildfire events in the database, monthly maps at 1-km resolution of area burned across the full western US domain, and, for each event, a shapefile with the known or estimated fire perimeter as well as a 1-km resolution map of the fraction each grid cell that burned."

Line 96: the different class of severity is presented here, it is not clear where these come from and if the classification is retained for further use. If these are all the range of possible classes provided by MTBS, and are no longer used, I recommend removing these as it will confuse the reader expecting such a classification.

We agree that mentioning the MTBS severity classifications is unnecessary here since we did not use them. We revised to simply say we calculate the area burned by MTBS fires as sum of area within 30-m grid cells classified as burned by MTBS.

Line 157: 200 kms is a large distance for exclusion, what is the rational for it and the impact of choosing a smaller distance.

This comment refers to a distance threshold we used to identify duplicate entries of the same fire in a fire database: when two fires have the same name (excluding fires of UNKNOWN name), start dates within 5 days, and ignition locations within 200 km, we determined these to be duplicates. We agree 200 km is a large distance and have revised the threshold down to 100 km. In this light, we also revised a previous criterion for linking non-MTBS fires to MTBS fires with matching names. We previously made linkages if a non-MTBS fire occurred within 1.5° of the bounding box of an MTBS fire (in addition to having a start date within 14 days), and we have revised this down to 1.0°.

Ultimately this decision about distance threshold is not majorly consequential. First, when two datasets, or the same dataset, lists a fire of the same name within days of each other, then the locations associated with these are generally near each other. For example, in our preparation of the FPA FOD dataset of >17,000 western US fires ≥1 km², using a threshold of 200 km previously led to an identification of 33 pairs of fires with identical names that were close enough in time and space to be treated as duplicates. Using the new threshold of 100 km reduces this number to 27. Among these 27 pairs, all but 6 are pairs are within 25 km of each other, and the 6 father than 25 km appear highly likely based on closer inspection to be true duplicates. As for the 6 pairs originally flagged as duplicates with a 200 km threshold, but not flagged with a 100 km threshold, it appears somewhat less certain that these are true duplicates. Three of these duplicate pairs do not have identical dates, for example, differences in fire size tend to be larger, and the shared fire names are somewhat more common. After describing the 100 km / 5 days criteria for treating identically named fires as duplicates, we now add (L162-165):

"These thresholds were determined empirically to allow for the automatic detection of the vast majority of duplicate fires with identical names. We found that loosening the distance and start-date criteria increasingly led to automated detections of duplicates that did not stand up to scrutiny, as pairs of identically named fires that are distant in time or space are more likely to have large discrepancies in reported fire sizes or to have commonly used fire names."

Importantly, similar to our response to the Reviewer's more general comment about the rules we use to detect duplicates, it is impossible to identify duplicate entries with 100% confidence and no set of thresholds used in our quality-control process can be perfect. This is why it is crucial that we perform rigorous manual screening of the datasets for duplicate or erroneous entries throughout the process of producing the WUMI. This includes countless visual inspections of fire lists as well as inspection of fire maps to identify instances where burned area accumulates in a given area within an unreasonably short time. Ultimately the distance and time thresholds we use to identify duplicate fires are not highly influential on the final dataset because of the rigor with which we manually pore over the data to minimize errors. We believe we have clarified this broader point through the text that we added to the end of section 2.2.2 (Dataset merging), which we also provided above in response to the Reviewer's general comment.

Line 162-163: I assume by "keep the first fire in the database" it is meant retaining the record with the earliest date. If so, please make it clear.

The text the reviewer refers to intends to say that when two or more fires have identical sizes and dates within a given database, such that they cannot be sorted by date, we retain just the first occurrence of that fire event in the database. We revised the text to clarify (L170-171):

"When fires within a given database have identical date and size, we retain only the first database instance."

Line 330: I might have missed but it is not clear what ST/C&L stands for This abbreviation stands for state, county, and local agencies. The abbreviation was and still is defined on the previous line.

---

## Author Comment (AC2)

**Reviewer 2**

Williams et al. introduce WUMI2024a, a harmonized, quality-controlled database of 22,464 large wildfire events across the 11 western U.S. states from 1984 – 2024. The product merges seven public government datasets and provides, for each event, a start date, location, final size, a perimeter (observed where available, circular when missing), and a 1-km grid of fractional area burned. About 46% of events inherit 30 m burned-area maps and perimeters from MTBS; another 24% have perimeters from non-MTBS sources; only 30% lack observed perimeters. Given the importance and lack of good quality data for wildfire regime assessment, the work of Williams et al. is highly relevant and useful for the community.

We thank the reviewer for their thorough and constructive review of our paper. We address each comment below (in blue font). In addressing the reviews we discovered some minor errors in our code and also performed additional manual inspections of the fire lists and burned area maps that led to a revised dataset with approximately 1% fewer wildfires. This revision did not cause notable changes to the regional patterns of fire frequency or burned area. The paper and the WUMI2024a database archived online have been revised to reflect these changes.

**Major comments:**

1. The paper introduces an alternate event list that replaces MTBS "parent" fires with linked non-MTBS "sub-fires," and even reassigns sizes so sub-fires sum to the parent (Sec. 2.4). It is not clear what decision logic was used to separate single events vs. multiple events. How are overlaps across days/IDs handled? What are the implications for trend analysis of counts vs. burned areas when switching between parent and sub-fire lists?

This comment and set of questions motivated us to expand in section 2.4 (Summary of final dataset) on how sub-fires were treated. We describe the text addition below, but first we'd like to clarify to the reviewer that sub-fires were identified using the basic methods we developed to identify linkages between MTBS and non-MTBS fires, described in the 3rd paragraph in section 2.2.2 (Dataset merging). That is, a 'parent' fire is identified as any MTBS fire for which we found multiple linkages to non-MTBS fires. In these cases, the multiple non-MTBS fires linked to the same parent MTBS fire are called sub-fires. We have now added a sentence to section 2.2.2 that indicates to the reader that a description will be provided in section 2.4 regarding how these cases of multiple sub-fires being part of the same parent fire are handled. We also revised section 2.4 to better clarify:

First, at the beginning of section 2.4 immediately after indicating that a number of MTBS fires are composed of multiple non-MTBS fires, we have added a sentence that we feel will clarify to a reader what we are referring to:

"For example, some fire complexes, when multiple fires that began independently but merged, are represented by MTBS as single events."

More generally, this reviewer comment motivated us to thoroughly revise section 2.4 to better describe the logic behind our treatment of sub-fires and the implications. After explaining that we adjust the ignition locations and fire sizes of sub-fires to be consistent with the burned-area maps of the parent fires, we have added (L274-283):

"We realign the sizes and locations of sub-fires to be consistent with the MTBS parent because the MTBS maps of area burned are satellite-based and rigorously quality controlled. In contrast, the fire sizes and ignition locations reported in other datasets are often approximate, as exemplified by cases in which the same fire is reported by different agencies but with non-identical sizes and ignition locations. On the other hand, adjustment of fire size and ignition location introduces errors for cases in which one or more of the sub-fires was misidentified (did not actually contribute to the burned area mapped by MTBS), not all sub-fires were identified, or two or more sub-fires were actually the same event but not identified as such. That said, we find that, among the 119 cases in which a parent fire is composed of sub-fires, the sum of the areas burned by the sub-fires consistently agree well with the total area of the parent fire (Fig. S1), giving us confidence that any unintended errors caused by adjustments to sub-fire sizes and ignition locations are generally minor."

In the above text we now also reference a new Supplementary Figure S1, which we provide below, that shows that the sums of the sizes of sub-fires agree very well with the sizes of their associated parent fires, which serves as evidence that our method of matching parent fires with their associated sub-fires works well.

Figure s1. Scatter plot comparing the sums of sub-fire sizes to the sizes of their corresponding parent fires. Dots represent each of 119 cases in which a single MTBS parent fire was found to be composed of two or more non-MTBS sub-fires. The diagonal black line is a one-to-one line.

We then go on to more thoroughly describe cases in which consideration of sub-fires would be desirable over parent fires and vice versa, clarify that these instances of parent and sub-fires are relatively rare and thus will not majorly impact analyses at the large scale of the western US, and also assure the reader that the WUMI2024a also includes the pre-adjustment sizes and locations of sub-fires (L285-294):

"Thus, the WUMI2024a includes two lists of wildfire events. The list that prioritizes sub-fires over parent fires may be preferable when the goal is to use the most accurate records of ignition dates and fire frequencies. However, perimeter data are not available for some sub-fire events, and MTBS is the only dataset that provides maps of burned versus unburned areas within the fire perimeter. Therefore, prioritization of parent fires over sub-fires may be preferable in

applications requiring the most accurate maps area burned. Importantly, the 119 parent fires and their 413 sub-fires are rare, constituting

Figure 1. Flowchart outlining the approach to develop the WUMI2024a.

4. The manuscript carefully notes under-representation of pre-1992 non-forest fires and potential artifacts post-2020. Based on these weaknesses, could you add a paragraph/section that outlines "best practices" and makes recommendations for potential users?

Thank you for this suggestion. In addition to deepening our discussion of the under-representation of pre-1992 fires in non-forest areas in section 3 (Data quality and dataset intercomparison), which includes a new state-by-state assessment of how well WFAIP agrees with FPA FOD during 1992–2001 and an associated supplementary figure (Fig. S3), we have added a paragraph that summarizes the dataset weaknesses and our associated recommendations to the Conclusions section. The new paragraph reads (L489-509):

"The WUMI2024a also has caveats. First, it relies on publicly available government records of wildfire incidents, which can be more accurately characterized as records of fire response and management rather than as purely records of wildfire occurrence. This means the WUMI2024a is missing any wildfires not recorded by a fire management agency and incorporated into a publicly available database. For example, some wildfires may have never been detected and reported, and others may have been extinguished without receiving official attention from a fire management agency. Thus, the database's completeness and accuracy are subject to temporal and geographic inconsistencies related to changes in practices related to fire detection, management, and data archival. These limitations are well exemplified by the fact that the

WUMI2024a indicates zero wildfires in Wyoming and New Mexico in 1984 (Fig. S2). The lack of large 1984 wildfires in two western US states is likely an artifact and we suggest excluding 1984 in assessments that include the interior Western US. We further presented evidence that fire frequencies in the WUMI2024a are highly likely to be artificially low prior to 1992 (the first year of the FPA FOD) in non-forested areas of the interior West because many wildfires in these areas are addressed by non-federal agencies (ST/C&L), which are not represented in the WFAIP database (Figs. 4, S3a). The negative biases in pre-1992 frequencies are likely to be largest in the non-forest areas of New Mexico, Arizona, Colorado, and Wyoming, as these are the areas where ST/C&L fires are most common in the FPA FOD, so we caution against analyses that rely on consistent fire reporting from the 1980s through 1990s in these areas. Our database does comprehensively represent non-federal fires in California, on the other hand, via data from CalFire. Finally, we were unable to find perimeter data for 29% of the wildfire events in the WUMI2024a. While these fires are relatively small, only accounting for less than 5% of area burned in the dataset, and we do adjust the circular perimeters and maps of burned area to exclude open water, ice, and barren ground, we warn against use of the circular fire features for applications that require accurate maps of area burned. These derived features are provided as resources for applications where an approximated map of area burned is better than none at all (e.g., as inputs for some hydrological or smoke-emission modelling exercises), but should be used with caution."

**Minor comments:**

1. The abstract cites 22,464 events, while Sec. 2.4 first lists 22,228 parent events and then 22,464 with sub-fire replacement. Could you clarify where the difference comes from? The difference is that often times fires represented by MTBS as individual events are actually composed of multiple smaller fires. In some cases these are official fire complexes in which multiple smaller fires merged to become a single larger fire, and in other cases what the MTBS team mapped as single event based on satellite imagery was actually constituted of multiple individual events. For example, MTBS fires often consist of several disconnected burned areas and sometimes these disconnected burned areas are distinguished by other data sources as independent fire events. In section 2.4 we specify that in cases like the ones described above we refer to the large burned area represented by MTBS as a 'parent' event and to the smaller events identified as within that parent event as sub-fires.

Thus, the larger number of events indicated in the original submission (22,464) was the total number of fires when parent fires are replaced by their sub-fires and the smaller number (22,228) represented the number of fires when parent fires are considered in place of their sub-fires. Note that these numbers have changed slightly, as identification of a small coding error and additional quality controlling reduced the number of fires by  $\sim$ 1%.

Importantly, we did describe the difference behind these two numbers in the first paragraph of section 2.4, which is the section that the reviewer refers to above. We have now revised that section to more clearly describe the difference between the two sets of fires representing differing numbers of events as well as why one dataset may be preferrable over another depending on the goal of the analysis. The new text is provided above in our response to this reviewer's first major comment.

2. L180 - 185 Where ignition points fall just outside the CalFire perimeter and are "snapped" to the nearest boundary, mention whether snapping is recorded as a flag and provide displacement statistics (median/95th percentile).

This comment refers to our method of adjusting ignition locations for some FPA FOD or WFAIP fires that are linked to a CalFire fire. Specifically, when a FPA FOD or WFAIP is linked to a CalFire fire, but the ignition location for the FPA FOD or WFAIP fire falls somewhere outside the perimeter of the CalFire fire (which is expected on occasion because records of ignition location are often imprecise), we adjust the FPA FOD or WFAIP ignition location to the nearest point along the CalFire perimeter. While this reviewer comment is specific to CalFire perimeters, we also perform this adjustment for FPA FOD and WFAIP fires that are linked to non-CalFire perimeters (MTBS, WFIGS, USGS, IAFPH). This was noted in section 2.2.3 about merging with supplemental datasets that include fire perimeters, but we failed to indicate that this is also done when we merge the non-MTBS fires with MTBS fires so we have added a sentence to the end of the paragraph in section 2.2.2 that describes merging non-MTBS fires with the MTBS dataset.

Importantly, instances of altered ignition locations for FPA FOD and WFAIP fires are relatively rare, and when they occur the adjustments are minor. The adjustments are rare because usually the ignition locations reported in the FPA FOD and WFIAP datasets already exist within the perimeters of the fires they are linked to. Of all 9,434 FPA FOD or WFAIP fires that appear in the final dataset, including sub-fires, 1,585 (16.8%) had adjusted ignition locations. Of these, the median adjustment distance was 406 m and 95% of distances were ≤6.6 km.

We have added a short paragraph to section 2.4 (Summary of the final dataset) describing this and how interested users can access the pre-adjusted ignition coordinates (L295-303):

"As mentioned above, the final ignition locations for some FPA FOD and WFAIP fires are different from those originally reported because, when we found a linkage between an FPA FOD or WFAIP fire and a fire from an alternative dataset that provides perimeter data, we adjusted FPA FOD or WFAIP ignition locations to the nearest point along the perimeter if the originally reported ignition location fell outside the perimeter. Notably these adjustments were relatively rare and minor. Of the 9,434 FPA FOD or WFAIP fires in the final dataset, ignition locations are adjusted for 16.8%. Of these, the median adjustment distance is 406 m and 95% of adjustments were within 6.6 km. Users can retrieve the originally reported ignition locations from the original lists of FPA FOD and WFAIP fires that we provide as part of the WUMI2024a archive by cross-referencing the unique FPA FOD and WFAIP fire identifiers provided in the final lists of WUMI2024a fires with those in the original lists."

---

## Referee Report (RR1)

Thank you to  the authors for their replies and revision of the manuscript. From my point of view, the revised version is ready for publication.

---

## Author Response (AR2)

Dear editor,

Here we submit the final version of our accepted manuscript. We were asked to thoroughly check over the text before submitting the final version. In doing so we found a number of typos which we fixed, and we also made additional minor revisions to clarify the language in some places, but our edits never altered the meaning of the text. Below this letter we have provided a PDF of the track-changes version of the final manuscript so you can easily confirm that our edits did not alter the substance of the paper.

Thank you very much for your time and care in handling this paper.

Sincerely,

Park Williams and co-authors

[revised manuscript text omitted]

235 in the satellite imagery by the MTBS team. All remaining MTBS fires without linkages to non-MTBS fires are added to the list of non-MTBS fires to complete our final list of WUMI2024a wildfire events. When linking FPA FOD and WFAIP fires to MTBS fires, we adjust any FPA FOD and WFAIP ignition location that falls outside its associated MTBS polygon to lie on the nearest point along the MTBS polygon.

240 Importantly, we found it infeasible to automate a one-size-fits-all set of rules that effectively detects duplicate fires within or between datasets without need for intensive additional scrutiny. For example, the same fire is often assigned different names by different government agencies; the same name may be spelled or misspelled in a range of ways; and dates, locations, and sizes are not always reported consistently across agencies, sometimes with large differences. Therefore, at every step of the quality-control and dataset-merging process we conduct rigorous visual inspections of fire lists and perimeter maps to identify

245 additional duplicate fires between or within datasets as well as fires that were misidentified as redundant by our automated routine. Due to the intensive nature of the visual inspection and manual correction process we are confident that the quality of

the final WUMI2024a dataset, while certainly not without remaining errors, is not highly sensitive to the specific rules implemented in the automated portion of the process.

**2.2.3 Supplementary datasets with fire perimeters**

Although WFIGS is incomplete prior to 2021, USGS does not provide ignition locations or records of fires lacking perimeter data, and IAFPH does not provide ignition location or start date, these datasets can provide unique geographic information for many fires in the WUMI2024a. We therefore treat WFIGS during 2018–2020, USGS, and IAFPH as supplementary. During each supplementary dataset's period of overlap with the WUMI2024a (2018–2020 for WFIGS, 1984–2020 for USGS, 1984–2023 for IAFPH), we identify matching fires in the WUMI2024a. Matches are identified when names match, fire starts are within 14 days (or same year in the case of IAFPH), and the WUMI2024a ignition location is within 0.25° of the supplementary perimeter's bounding box. In cases of matching UNKNOWN fire names, the WUMI2024a ignition location must be within the supplementary fire's bounding box and its size must be within 10% of the supplementary fire's size. If a WUMI2024a fire is matched to more than one supplementary fire, WFIGS is prioritized because it provides ignition locations and start dates, then USGS because it provides start dates, and then IAFPH. In the case of a match to an FPA FOD or WFAIP fire, that fire is assigned the perimeter and size of the supplementary fire and the ignition location is adjusted if outside the fire perimeter. In the case of a match between WFIGS and CalFire, the WFIGS ignition location is assigned to the CalFire fire and adjusted if outside the CalFire perimeter. Matches to other datasets (e.g., a supplementary fire to an MTBS fire) are recorded but no changes are made. Finally, for any FPA FOD or WFAIP fire not assigned a perimeter from an alternative dataset, we initially assume the fire perimeter to be circular centred on the ignition location. We then adjust the boundaries and sizes of circular perimeters such that burned areas exclude areas of open water, permanent ice, or barren ground according to the previous year's map of landcover type from the USGS annual National Land Cover Database (NLCD) (USGS, 2024).

**2.3 Burned-area maps**

For each fire in the WUMI2024a we produce a 1-km resolution map of fractional area burned. For MTBS fires, this is done by, for each 1-km grid cell in our study region, summing the area, in units of $km^2$, of all 30-m grid cells identified as burned in the original MTBS dataset. For fire perimeters from CalFire, WFIGS, USGS, and IAFPH, we assume the area within the fire perimeter is burned uniformly, accounting for partial cell overlap with the fire perimeter as well when fire perimeters indicate gaps within the area burned or disconnected burned areas. Notably, many of these non-MTBS perimeters do include within-fire gaps where burning did not occur. For FPA FOD and WFAIP fires assumed to be circular, we adjust the 1-km grids of area burned by multiplying by the fractional land cover that is not open water, permanent ice, or barren ground according to the NLCD, thus excluding areas deemed unburnable. Finally, as a quality control measure we make regional maps of monthly and annual area burned to identify cases of suspiciously high burned areas concentrated in time. This leads us to find duplicate fires not previously identified and, in some cases, use internet searches to identify incorrect dates or locations of fires. Over several iterations we correct found issues and remake the dataset following the methods described above.

**2.4 Summary of the final dataset**

The WUMI2024a database represents 21,940 wildfires ≥1 km² in size according to the area within the fire perimeter if MTBS fires are always prioritized as single events even if they are composed of multiple smaller events. However, 119 MTBS fires are composed of multiple non-MTBS fires. For example, some fire complexes, defined as fire events composed of multiple

290 fires that began independently but merged, are represented by MTBS as single events. In these cases, we refer to the large MTBS fires as parent fires and the smaller fires within as sub-fires. Because these sub-fires generally have different ignition locations and start dates from the parent fire, we produce an alternative list of WUMI2024a wildfire events that replaces each parent fire with its sub-fires. In this version, we adjust the sizes of sub-fires so that they sum to equal the sizes of the parent fires. When the ignition location for a sub-fire is not within the satellite-derived area burned of the MTBS parent fire, we adjust

295 the ignition location to the nearest burned location within the parent fire's footprint. In the alternative list of wildfire events that replaces each parent fire with its sub-fires, there are 22,234 wildfires ≥1 km² in size. We realign the sizes and locations of sub-fires to be consistent with the MTBS parent because the MTBS maps of area burned are satellite-based and rigorously quality controlled. In contrast, the fire sizes and ignition locations reported in other datasets are often approximate, as exemplified by cases in which the same fire is reported by different agencies but with non-identical sizes and ignition locations.

300 On the other hand, adjustment of fire size and ignition location introduces errors for cases in which one or more of the sub-fires was misidentified (did not actually contribute to the burned area mapped by MTBS), not all sub-fires were identified, or two or more sub-fires were actually the same event but not identified as such. That said, we find that, among the 119 cases in which a parent fire is composed of sub-fires, the sum of the areas burned by the sub-fires consistently agree well with the total area of the parent fire (Fig. S1), giving us confidence that any unintended errors caused by adjustments to sub-fire sizes and

305 ignition locations are generally minor.

Thus, the WUMI2024a includes two lists of wildfire events. The list that prioritizes sub-fires over parent fires may be preferable when the goal is to use the most accurate records of ignition dates and fire frequencies. However, perimeter data are not available for some sub-fire events, and MTBS is the only dataset that resolves unburned areas within fire perimeters.

310 Therefore, prioritization of parent fires over sub-fires may be preferable in applications requiring the most accurate maps of area burned. Importantly, the 119 parent fires and their 413 sub-fires are rare, constituting <2% of all fires in the dataset, meaning that analyses of fire frequencies or areas burned at the large scale of the western US will not be strongly sensitive to whether parent fires or sub-fires are prioritized. Additionally, for each parent fire the WUMI2024a database includes a text file that provides the pre-adjustment ignition locations and sizes of sub-fires. Table S1 provides descriptions of the wildfire

315 attributes included in the WUMI2024a lists of wildfires and sub-fires.

As mentioned above, the final ignition locations for some FPA FOD and WFAIP fires are different from those originally reported because, when we found a linkage between an FPA FOD or WFAIP fire and a fire from an alternative dataset that

provides perimeter data, we adjusted FPA FOD or WFAIP ignition locations to the nearest point along the perimeter if the originally reported ignition location fell outside the perimeter. Notably these adjustments were relatively rare and minor. Of the 9,434 FPA FOD or WFAIP fires in the final dataset, ignition locations are adjusted for 16.8%. Of these, the median adjustment distance is 406 m and 95% of adjustments were within 6.6 km. Users can retrieve the originally reported ignition locations from the original lists of FPA FOD and WFAIP fires that we provide as part of the WUMI2024a archive by cross-referencing the unique FPA FOD and WFAIP fire identifiers provided in the final lists of WUMI2024a fires with those in the original lists of fires represented in those databases.

[revised manuscript text omitted]
. Assessing by state, the negative bias in pre-1992 fire frequency is likely to be strongest in New Mexico, where, during 1992–2001, the frequency of non-forest fires ≥1 km$^2$ was nearly 70% lower according to WFAIP than to FPA FOD and, non-coincidentally, ST/C&L fires accounted for nearly 70% of the FAP FOD fire frequency (Fig. S3a). Negative biases in WFAIP fire frequencies are also relatively large, on the order of 30–40%, in non-forest areas of Arizona, California, Colorado, and Wyoming. These biases are also attributable mostly to the absence of ST/C&L fires. In the California, however, the negative bias in WFAIP fire frequency is largely addressed in the WUMI2024a through use of CalFire data, which do include fires addressed by non-federal agencies. Thus, the WUMI2024a is likely to under-represent the frequencies of pre-1992 wildfires in non-forest areas, particularly in New Mexico, Arizona, Colorado, and Wyoming. In addition, the maps of annual fire occurrences provided in Fig. S2 indicate a complete lack of fires in 1984 in the states of Wyoming and New Mexico, which is almost certainly an artifact of inconsistent reporting practices.

435  Importantly, the bias toward too few pre-1992 fires is less consequential to assessments of area burned than to fire frequency, as indicated by a much better match between the FPA FOD and WFAIP time series of area burned during their period of overlap (Fig. 5; Fig. S3b). This is because the ST/C&L fires missing from WFAIP tend to be relatively small, whereas regional time series of area burned are dominated by a small proportion of very large fires that more often command federal responses, most of which appear in the MTBS dataset. Users who prefer a dataset built from a more temporally consistent set of data

440  sources may consider excluding FPA FOD fires reported by ST/C&L agencies. Overall, the WUMI2024a underrepresents fire frequencies in 1984–1991 relative to 1992–2020, but this effect appears isolated mostly to relatively small fires in non-forest areas, particularly in the easternmost states of the western US. We therefore caution against over-interpretation of the relatively low frequency of pre-1992 fires, particularly small non-forest fires, as represented by WUMI2024a or other databases that extend back to the 1980s or prior.

445

There are probably additional artifacts associated with the 2020-to-2021 transition from reliance on FPA FOD to WFIGS as the primary source of wildfire occurrence data. While WFIGS does include fires reported by non-federal agencies, fires are not included in WFIGS if strict reporting standards are not met. The effects of this shift are difficult to readily assess because WFIGS is not complete during its 2018–2020 period of overlap with FPA FOD. Some insights may be gleaned, however,

450  through comparison to the independent satellite-based FIRED dataset (Balch et al., 2020), a delineation of the gridded MODIS v6.1 burned area product (Giglio et al., 2016) into individual fire events. The blue dashed time series in Fig. 4 show MODIS-derived annual fire frequencies, excluding fires <1 $km^2$ in size (assuming burned 500-m MODIS grid cells burn uniformly) as well as fires in heavily urbanized or agricultural areas where satellite fire detections are less likely to represent wildfires. Comparing FIRED to WUMI2024a in non-forest areas, we see WUMI2024a fire frequencies tend to be somewhat higher than

455  those derived from FIRED in 2001–2020, but not during 2021–2023 (Fig. 4, right side). This suggests that WFGIS may systematically underrepresent fire frequencies relative to what would be expected if FPA FOD was available following 2020. In forest areas, on the other hand, FIRED fires are consistently more frequent than indicated by WUMI2024a (Fig. 4, left side). The higher forest-fire frequencies in FIRED are most prominent in PNW and N Rockies, but also emerge in CA/NV and 4

Corners after 2017. However, there is not a clear post-2020 shift in WUMI2024a forest-fire frequencies relative to those

460 calculated from FIRED and it is therefore unclear how the shift from FPA FOD to WFIGS may affect accuracy of post-2020

forest-fire frequencies.

The higher frequencies of forest fires in the MODIS FIRED dataset relative to WUMI2024a are interesting. One likely cause

is that government records often attribute geographically separated burned areas to the same event, but the FIRED dataset may

465 more often delineate these as distinct events. In addition, the FIRED dataset does not distinguish prescribed fires from wildfires

and may also capture wildfires in remote areas if the fires are small and short-lived enough to not be managed by a federal or

local agency. Importantly, records of annual burned area derived from the FIRED dataset are in tight agreement with the

[revised manuscript text omitted]

The WUMI2024a also has caveats. First, it relies on publicly available government records of wildfire incidents, which can be more accurately characterized as records of fire response and management rather than as purely records of wildfire occurrence. This means the WUMI2024a is missing any wildfires not recorded by a fire management agency and incorporated into a

530 publicly available database. For example, some wildfires may have never been detected and reported, and others may have been extinguished without receiving official attention from a fire management agency. Thus, the database's completeness and accuracy are subject to temporal and geographic inconsistencies related to changes in practices related to fire detection, management, and data archival. These limitations are well exemplified by the fact that the WUMI2024a indicates zero wildfires in Wyoming and New Mexico in 1984 (Fig. S2). The lack of large 1984 wildfires in two western US states is likely an artifact

535 and we suggest excluding 1984 in assessments that include the interior Western US. We further presented evidence that fire frequencies in the WUMI2024a are highly likely to be artificially low prior to 1992 (the first year of the FPA FOD) in non-forested areas of the interior West because many wildfires in these areas are addressed by non-federal agencies (ST/C&L), which are not represented our database prior to 1992 (Figs. 4, S3a). The negative biases in pre-1992 frequencies are likely to be largest in the non-forest areas of New Mexico, Arizona, Colorado, and Wyoming, as these are the areas where ST/C&L

540 fires are most common in the FPA FOD, so we caution against analyses that rely on consistent fire reporting from the 1980s through 1990s in these areas. Our database does comprehensively represent non-federal fires in California, on the other hand, via data from CalFire. Finally, we were unable to find perimeter data for 29% of the wildfire events in the WUMI2024a. While

545 these fires are relatively small, only accounting for less than 5% of area burned in the dataset, and we do adjust the circular

perimeters and maps of burned area to exclude open water, ice, and barren ground, we warn against use of the circular fire

features for applications that require accurate maps of area burned. These derived features are provided as resources for

applications where an approximated map of area burned is better than none at all (e.g., as inputs for some hydrological or

smoke-emission modelling exercises), but should be used with caution.

550

Finally, the value of the WUMI2024a can be improved upon by expanding it to include more than wildfire start dates, locations,

causes, perimeters, and 1-km resolution maps of area burned. For example, many fires in the WUMI2024a can be linked to

those represented in the ICS-209-PLUS database (St. Denis et al., 2023) to provide information about fire management costs,

personnel, and impacts on people and property. Additionally, Landsat imagery can be used to improve our maps of area burned

555 for fires in the WUMI2024a that are not represented among the fires $>4.04$ km$^2$ in size mapped by the MTBS. Landsat imagery

can also be used to detect the perimeters and areas burned for fires with unknown perimeters that we currently assume to be

circular, identify additional fire events not yet represented in our database, and expand beyond our maps of burned area to also

develop high-resolution maps of fire severity (e.g., Parks et al., 2019) for all fires in the database. Likewise, the MODIS-based

FIRED dataset (Balch et al., 2020) can be used to decompose the fire maps in our database into maps of daily fire progression,

560 and capabilities for mapping of fire spread are deepening further through use of the Visible Infrared Imaging Radiometer Suite

(VIIRS) instrument (Schroeder et al., 2014; Chen et al., 2022). Imagery from MODIS and VIIRS can also be used to map fire

intensity via retrievals of fire radiative power (Schroeder et al., 2014; Giglio et al., 2016). While the WUMI2024a database of

wildfire events has been designed to accommodate future updates and expansions, including additional metrics related to fire

processes (e.g., spread) and impacts (e.g., severity), its current form advances beyond currently available databases of western

565 US wildfire events and will be a robust and valuable resource for researchers and practitioners in the field of wildfire science.

**6 Author contributions**

[revised manuscript text omitted]

St. Denis, L. A., Short, K. C., McConnell, K., Cook, M. C., Mietkiewicz, N. P., Buckland, M., and Balch, J. K.: All-hazards dataset mined from the US National Incident Management System 1999–2020, Scientific data, 10, 112, https://doi.org/10.1038/s41597-023-01955-0, 2023.

USGS: Annual National Land Cover Database (NLCD) Collection 1.0, https://doi.org/10.5066/P94UXNTS, 2024.

Welty, J. L. and Jeffries, M. I.: Combined wildland fire datasets for the United States and certain territories, 1800s-Present, https://doi.org/10.5066/P9ZXGFY3, 2021.

Westerling, A. L.: Wildfire Simulations for California's Fourth Climate Change Assessment: Projecting Changes in Extreme Wildfire Events with a Warming Climate, California Energy Commission, 2018.

Westerling, A. L., Turner, M. G., Smithwick, E. A. H., Romme, W. H., and Ryan, M. G.: Continued warming could transform Greater Yellowstone fire regimes by mid-21st century, Proceedings of the National Academy of Sciences USA, 108, 13165–13170, https://doi.org/10.1073/pnas.1110199108, 2011.

Williams, A. P. and Abatzoglou, J. T.: Recent advances and remaining uncertainties in resolving past and future climate effects on global fire activity, Current Climate Change Reports, 2, 1–14, https://doi.org/10.1007/s40641-016-0031-0, 2016.

Williams, A. P., Livneh, B., McKinnon, K. A., Hansen, W. D., Mankin, J. S., Cook, B. I., Smerdon, J. E., Varuolo-Clarke, A. M., Bjarke, N. R., and Juang, C. S.: Growing impact of wildfire on western US water supply, Proc Nat Acad Sci USA, 119, e2114069119, https://doi.org/10.1073/pnas.2114069119, 2022.

Williams, A. P., Juang, C. S., and Short, K. C.: The western united States MTBS-Interagency database of large wildfires, 1984–2024 (WUMI2024a) (2024a), https://doi.org/10.5061/dryad.63xsj3vd4, 2025.